# Temperature Dependence of the Thermo-Optic Coefficient of GeO_2_-Doped Silica Glass Fiber

**DOI:** 10.3390/s24154857

**Published:** 2024-07-26

**Authors:** Gaspar Mendes Rego

**Affiliations:** 1ADiT-LAB, Instituto Politécnico de Viana do Castelo, Rua Escola Industrial e Comercial Nun’Álvares, 4900-347 Viana do Castelo, Portugal; gaspar@estg.ipvc.pt; 2Center for Applied Photonics, INESC TEC, Rua Dr. Roberto Frias, 4200-465 Porto, Portugal

**Keywords:** silica glass, refractive index, material dispersion, thermo-optic coefficient, cryogenic temperatures

## Abstract

In this paper we derived an expression that allows the determination of the thermo-optic coefficient of weakly-guiding germanium-doped silica fibers, based on the thermal behavior of optical fiber devices, such as, fiber Bragg gratings (FBGs). The calculations rely on the full knowledge of the fiber parameters and on the temperature sensitivity of FBGs. In order to validate the results, we estimated the thermo-optic coefficient of bulk GeO_2_ glass at 293 K and 1.55 μm to be 18.3 × 10^−6^ K^−1^. The determination of this value required to calculate a correction factor which is based on the knowledge of the thermal expansion coefficient of the fiber core, the Pockels’ coefficients (*p*_11_ = 0.125, *p*_12_ = 0.258 and *p*_44_ = −0.0662) and the Poisson ratio (*ν* = 0.161) of the SMF-28 fiber. To achieve that goal, we estimated the temperature dependence of the thermal expansion coefficient of GeO_2_ and we discussed the dispersion and temperature dependence of Pockels’ coefficients. We have presented expressions for the dependence of the longitudinal and transverse acoustic velocities on the GeO_2_ concentration used to calculate the Poisson ratio. We have also discussed the dispersion of the photoelastic constant. An estimate for the temperature dependence of the thermo-optic coefficient of bulk GeO_2_ glass is presented for the 200–300 K temperature range.

## 1. Introduction

Temperature impacts the performance of optical fiber communication systems, since the fiber itself and its properties: loss, dispersion and birefringence, … are affected by heat transfer, as well as fiber lasers, amplifiers and gratings [1,2,3]. Modeling the thermal behavior of fiber gratings in extreme conditions such as at cryogenic temperatures is very important in order to assess the performance of sensor devices [4,5,6]. Therefore, the precise knowledge of a key parameter such as the thermo-optic coefficient of fiber glass materials is on demand [7]. Recently [8], we have discussed the expected values for the thermo-optic coefficient, d*n*/d*T*, of silica glass, the main constituent of an optical fiber. In order to obtain d*n*/d*T* for the fiber core, we need to know its composition and bear in mind that during fiber drawing, due to the different viscosities of core and cladding materials, stresses will be induced and, therefore, the final refractive index will also be affected. Unfortunately, data concerning GeO_2_ glass is scarce and the methodologies used to extract data have their own limitations. For instance, the core’s thermo-optic coefficient could be determined if data concerned the refractive index of bulk GeO_2_ glass would be available for different temperatures and wavelengths, similarly to the work presented by Leviton and Frey for SiO_2_ [9]. Also, the work presented by Fleming [10] would solve the problem if performed at different temperatures. M. Medhat et al. [11] presented values but for the visible wavelength range and for a GeO_2_-doped silica glass fiber with a graded index profile. Another approach would be the use of Ghosh’s model [12,13], however the coefficients need to be adjusted for the SMF-28 fiber which may not be straightforward [8,14]. An estimative based on a pair of long period fiber gratings was also presented [15], but the value obtained for the GeO_2_-doped fiber seems to be high, even for the 30–60 °C temperature range. Fiber Bragg gratings (FBGs) were also employed to cover the 50–800 °C [16]. In this paper we present a methodology based on the knowledge of the fiber and on the temperature sensitivity of FBGs inscribed on it. The method also enables to extract the thermo-optic coefficient of the bulk material used for doping the fiber core. Although other core dopants can be considered, which imposes no limitation on the presented methodology, for the sake of simplicity and to control the dopants impact we shall focus on a well-known weakly-guiding fiber, the SMF-28 from Corning which have a GeO_2_ dopant concentration less than 4 mol%. FBGs imprinted in the SMF-28 fiber will be used to extract the temperature behavior of the effective refractive index that will be afterwards related to d*n*/d*T* of the fiber core through a set of equations derived for weakly-guiding fibers. Finally, we will use the additivity model to determine the thermo-optic coefficient of bulk GeO_2_ glass. Its temperature dependence will be also discussed through the knowledge of FBGs and the thermal expansion coefficient. To validate the obtained results, we will use data from other optical devices such as interferometers.

The paper is organized in three sections. The first is related to weakly-guiding fibers: its full characterization and the derivation of the equation that enables the determination of the core thermo-optic coefficient through the use of the additivity model. The second to fiber Bragg gratings: its effective refractive index, the temperature sensitivity and the determination of the correction factor. The third section is dedicated to the calculation of the correction factor for the SMF-28 fiber as a function of the Poisson ratio and Pockels’ coefficients. The analysis requires the correlation of data obtained by using the stimulated Brillouin gain spectrum, the acoustic velocities, whispering gallery modes, the elastic properties of the fiber, the photoelastic coefficient and the thermal expansion coefficient of the fiber. Finally, we validate the values obtained for the thermo-optic coefficient by comparing with published results achieved by using Fabry-Perot interferometers.

## 2. Ge-Doped Silica Glass Fiber

In order to properly model fiber gratings we need to characterize the host fiber, namely to have knowledge of the core diameter and of the refractive index difference between the core and cladding regions. The SMF-28 fiber from Corning is one of the most studied standard fibers and the reasons have been pointed out by researchers as being reliable with well-defined parameters and, therefore, having a wide application in optical communications and sensing [17,18,19]. However, different values can be found in the literature and even the datasheet should be used only as a reference [20,21,22,23]. Along the past 40 years several metrological standards have been used to measure the fiber parameters [24,25,26,27,28]. The best practice may recommend to collect the maximum information possible on the fiber and use waveguide equations to correlate the obtained parameters. For instance, the mode field diameter (*MFD*) at a particular wavelength and the cut-off wavelength (*λ*_c_) can be used to estimate the core radius (*D*_co_) though the Marcuse’s empirical formula [29,30,31,32]:(1)$$MFD=D_{co}\left[0.65+0.434{\left(\frac{\lambda }{\lambda_{c}}\right)}^{1.5}+0.015{\left(\frac{\lambda }{\lambda_{c}}\right)}^{6}\right],$$
and afterwards the numerical aperture (*NA*) can be determined by using the normalized frequency *V* expression:(2)$$V=\frac{\pi D_{co}}{\lambda }\sqrt{n_{co}^{2}-n_{cl}^{2}},$$
where *n*_co_ and *n*_cl_ are respectively, the refractive index of the fiber’s core and cladding. By putting *V* = 2.405 and *λ* = *λ*_c_ to yield:(3)$$NA=\sqrt{n_{co}^{2}-n_{cl}^{2}}=\frac{2.405\lambda_{c}}{\pi D_{co}}.$$

On the other hand, one may think that the refractive index profile (RIP) would clarify any potential ambiguity on the fiber parameters determination, but different techniques such as refracted near field (RNF), transmitted near field, transverse interferometric, quantitative phase imaging, reconstruction through tomographic stress measurement profiles, or using atomic force microscopy results in different values [21,33,34,35,36,37]. In fact, considering the most common technique (RNF), and by sweeping the fiber end-face at 0° and 90° may result in different values for Δ*n* and *D*_co_, being the difference larger for the latter. We should recall that fiber cleaving changes the stress distribution which also affects the refractive index [38] and the determination of its absolute value also requires calibration with a fluid of known refractive index. In general for the SMF-28 fiber, Δ*n* ranges from 4.4–5.4 × 10^−3^ and *D*_co_ from 8.0–8.8 μm and the most common values are: Δ*n* = 5.2–5.4 × 10^−3^ and *D*_co_ = 8.2–8.6 μm. Based on our measurements [39], we will consider Δ*n* ~ 5.4 × 10^−3^ and *D*_co_ ~ 8.6 μm which matches the values presented in [19]. Furthermore, the writing of weak FBGs in the SMF-28 fiber allows one to determine its effective refractive index (*n*_eff_) [40] and the obtained results are also consistent with the assumed fiber parameters. The SMF-28 is a weakly-guiding fiber [41] for which the normalized propagation constant can be written as:(4)$$b=\frac{n_{eff}-n_{cl}}{n_{co}-n_{cl}},$$
thus
(5)$$n_{eff}=n_{cl}+b\Delta n,$$
or be expressed as a function of the normalized frequency as [42]:(6)$$n_{eff}=n_{cl}+{\left(1.1428-\frac{0.996}{V}\right)}^{2}∆n.$$

Through the derivative in order to temperature results:(7)$$\frac{dn_{co}}{dT}=\frac{\frac{dn_{eff}}{dT}+2\left(1.1428-\frac{0.996}{V}\right)\frac{0.996}{V\lambda }\frac{d\lambda }{dT}\Delta n+\frac{dn_{cl}}{dT}\left[{\left(1.1428-\frac{0.996}{V}\right)}^{2}-1+\frac{2*0.996n_{cl}\left(1.1428-\frac{0.996}{V}\right)}{V\left(n_{co}+n_{cl}\right)}\right]}{\left[{\left(1.1428-\frac{0.996}{V}\right)}^{2}+\frac{2*0.996n_{co}\left(1.1428-\frac{0.996}{V}\right)}{V\left(n_{co}+n_{cl}\right)}\right]},$$

From the previous equation it can be concluded that one can determine the thermo-optic coefficient of the fiber core by knowing the fiber parameters, the thermo-optic coefficient of the fiber cladding (typically, pure-silica glass) and the temperature dependence of a fiber device, such as, a FBG. On the other hand, the thermo-optic coefficient of the fiber core is related to the thermo-optic coefficients of SiO_2_ and GeO_2_ by knowing the fractional volume of glass occupied by GeO_2_ (*m*) and using the additivity model [43]:(8)$$n_{co}={\left(1-m\right)n}_{SiO_{2}}+{mn}_{GeO_{2}},$$

(9)$$\frac{dn_{co}}{dT}=\left(1-m\right)\frac{dn_{SiO_{2}}}{dT}+m\frac{dn_{GeO_{2}}}{dT},$$
being *m* defined as:(10)$$m=\frac{\frac{M_{GeO_{2}}}{M_{SiO_{2}}}\;\;\frac{\rho_{SiO_{2}}}{\;\rho_{GeO_{2}}\;}\;x}{1+x\;\left[\frac{M_{GeO_{2}}}{M_{SiO_{2}}}\;\;\frac{\rho_{SiO_{2}}}{\;\rho_{GeO_{2}}\;}-1\right]},$$
and *x* is the molar fraction of GeO_2_ dopant concentration.

On contrary to pure silica, published values in the literature for the thermo-optic coefficient of pure GeO_2_ [44,45] is scarce and, therefore, this set of equations enables to determine its value, say at room temperature and for a particular wavelength (for instance, 293 K and 1.55 μm). At this point, we shall recall that the refractive index of the core and cladding materials may differ for the preform and for the optical fiber. The differences arise from mechanical and thermal stresses due to the different viscosity and thermal expansion coefficients of core and cladding materials and also from viscoelasticity, due to its time dependence induced during fiber drawing. Typically, the pure silica cladding bears the applied force and it has not enough time to reach thermodynamic equilibrium. The elastic stresses affect mainly the core, that is compressed by the cladding, while viscoelasticity affects mainly the pure silica cladding. Both contributes to a decrease of the refractive index of core and cladding materials in the fiber in comparison to the preform. Taking into account the value of 4.7 MPa [35], for the mean axial stress measured in the SMF-28, the cladding refractive index decrease due to frozen-in viscoelastic stress is calculated to be −3 × 10^−5^ [46] and can be, therefore, neglected. Furthermore, the residual elastic stresses contribute to a decrease in the cladding refractive index of about −2 × 10^−5^ and to an increase in the core of about 4 × 10^−5^ [47,48]. The overall contribution to Δ*n* is of the order of 1 × 10^−4^. On the other hand, it has been claimed that the refractive index of the cladding can be several parts in the 4th decimal place higher than that of annealed silica and that is attributed to quenching of the fiber during its production [24,28]. The reference value for annealed silica is the one obtained by Malitson [49] and it is known that the value obtained by Fleming for a quenched glass is about 3 × 10^−4^ higher [50]. However, as discussed in our previous paper [8], the values obtained by Leviton et al. for four samples of annealed silica glass are even higher than for quenched silica (for instance, the values for Corning 7980 silica sample are about 1 × 10^−4^ above) [9]. Gathering all the information concerning the fiber fabrication and the errors associated to the measurement of the refractive index profile, it is not possible to clearly state that the refractive index of the silica cladding is higher than for annealed bulk samples. Moreover, it is also known that values of the order of 300 g for the drawing tension can reduce the refractive index of the cladding by about 2 × 10^−4^ [51]. Returning to the SMF-28 fiber, for which only ~12.5 g (peak stress 10 MPa) are used for the drawing tension, we do not expect considerable changes in the cladding refractive index [38,52,53]. Furthermore, those changes are within the uncertainty of the measurements and in fact, more important than knowing the absolute value of the cladding refractive index is to know the index difference, Δ*n*.

As far as the core is concerned, each 1 mol% GeO_2_ accounts for 0.1% in Δ [54] and therefore for Δ*n* = 5.4 × 10^−3^ we expect ~3.7 mol% GeO_2_. Also, for the core we can question if we should use annealed or quenched GeO_2_. Calculations using the Sellmeier’s coefficients for pure GeO_2_ (quenched and annealed samples) presented in [55] and for silica, the ones obtained by Leviton and Frey for Corning 7980 [9], show that the difference in values for Δ*n* is of about 1 × 10^−4^ being the mol% of GeO_2_ 3.70 ± 0.04. Therefore, we will also use the GeO_2_ annealed sample, resulting in Δ*n* = 1.464 × 10^−3^ × [GeO_2_(mol%)] at 1.55 μm (valid for small concentrations since the dependence is in fact quadratic, Figure 1).

## 3. Fiber Bragg Gratings

The determination of the core thermo-optic coefficient requires the knowledge of the value of the effective thermo-optic coefficient. That, can be accomplished by following the thermal behavior of fiber Bragg gratings by using the temperature dependence of the Bragg wavelength, $\lambda_{B}$:(11)$$\lambda_{B}=n_{eff}\Lambda ,$$
where $n_{eff}$ is the effective refractive index and Λ the pitch of the phase-mask (which is twice that of the grating period).

The derivative of the grating resonance condition yields:(12)$$\frac{dn_{eff}}{dT}=n_{eff}\left(\frac{1}{\lambda_{B}}\frac{d\lambda_{B}}{dT}-\alpha_{cl}\right),$$
where $\alpha_{cl}\;$ represents the thermal expansion coefficient of the cladding material. It should be noted that being the cladding much larger than the core and since the thermal expansion coefficient of the core is larger than that of the cladding, it is the latter that defines the expansion of the grating period. On the other hand, the core is not free to expand and thus a compressive stress is induced in the core region during fiber heating leading to an increase of the refractive index. Therefore, the effective refractive index should be corrected by using the following expressions [56,57,58]:(13)$${\frac{dn_{eff}}{dT}}_{corr}=\frac{n_{eff}^{3}}{2}\left(2{\varepsilon oc}_{eff}+{\sigma oc}_{eff}\right)\left(\alpha_{co}-\alpha_{cl}\right),$$
where $\alpha_{co}\;$ represents the thermal expansion coefficient of the core material, that for the SMF-28 fiber can be calculated using the additivity model resulting in the following expression:(14)$$\alpha_{SMF28}=\frac{{\left(1-m\right)\;\rho }_{SiO_{2}}\;\alpha_{SiO_{2}}+{\;m\;\rho }_{GeO_{2}}\;\alpha_{GeO_{2}}}{\rho_{SMF28}},$$
and $\rho_{SMF28}\;$ is the density of the fiber core, determined as:(15)$$\rho_{SMF28}=\rho_{SiO_{2}}\left(1-m\right)+\rho_{GeO_{2}}m.$$

The ${\varepsilon oc}_{eff}$ and ${\sigma oc}_{eff}$ represent the strain and stress-optic effective coefficients and are expressed as:(16)$${\varepsilon oc}_{eff}=\left(mn_{GeO_{2}}^{3}{\varepsilon oc}_{GeO_{2}}+\left(1-m\right)n_{SiO_{2}}^{3}{\varepsilon oc}_{SiO_{2}}\right)\frac{1}{n_{eff}^{3}},$$

(17)$${\sigma oc}_{eff}=\left(mn_{GeO_{2}}^{3}{\sigma oc}_{GeO_{2}}+\left(1-m\right)n_{SiO_{2}}^{3}{\sigma oc}_{SiO_{2}}\right)\frac{1}{n_{eff}^{3}},$$
where *n*, *εoc*, and *σoc* represent the values of refractive index, strain, and stress-optic coefficients of the bulk materials:(18)$$\varepsilon oc=p_{12}-\nu \left(p_{11}+p_{12}\right),$$



(19)
$$\sigma oc=p_{11}-2\nu p_{12}.$$



The values of molar mass (*M*), density, thermal expansion coefficient [8], Pockels’ photoelastic coefficients (*p*_11_, *p*_12_) and Poisson ratio (*ν*) of germanium-doped silica glass are, respectively, presented in Table 1 [59].

There is a final issue requiring attention as a result of the fact that during the grating inscription a *δn*_co_ is induced in the fiber core (averaged over the grating length being half of the amplitude modulation for a grating with a duty-cycle of 0.5) which is larger for strong gratings as the reflectivity approaches 1. Consequently, the effective refractive index will also change. In this context, the induced effective refractive index *δn*_eff_ can be determined from the grating spectrum by knowing the Bragg wavelength, *λ*_B_ the grating length, *L* and the reflectivity, *R* and, therefore, *δn*_co_ can be afterwards obtained by using the confinement factor, *η* [60,61].
(20)$$\delta n_{eff}=\frac{\lambda_{B}}{2\pi L}{tanh}^{-1}\sqrt{R},$$



(21)
$$\delta n_{co}=\frac{\delta n_{eff}}{\eta },$$


(22)
$$\eta =1-\frac{1}{V^{2}}.$$



For the calculations we have assumed the values discussed in the previous section, namely, *D*_co_ = 8.6 μm and Δ*n* = 5.4 × 10^−3^ and for the cladding refractive index at 1.55 μm the value obtained for Corning 7980 (1.4444) [8], by using Equations (2) and (6) we determined the effective refractive index for the SMF-28 fiber. Afterwards, by considering a moderate grating (*R*~24%) [62], and by replacing the values in Equations (20)–(22) and (11), we estimated *δn*_eff_, *δn*_co_ and *Λ* to be respectively, 4.64 × 10^−5^, 5.96 × 10^−5^ and 1.0729 μm. Then, Equations (6)–(19) yields the following values for the thermo-optic coefficients (corrected effective, core and bulk GeO_2_): 8.45, 8.55 and 18.3 × 10^−6^ K^−1^. In order to validate our results we have also used a strong grating [63], although in this case we knew the pitch of the phase mask (1.070 μm) but we had to estimate the grating length since it was inscribed on the splice region of two dissimilar fibers being one of them the Corning SMF-28. Based on the knowledge that we had on the impact of the arc discharge on the fiber’s stress annealing (a region of about 1 mm) [64] and also on the separation of the peaks obtained in the Fabry-Perot spectrum (Δ*λ* = *λ*^2^/2*n*_co_*L* = 1 nm) [65] we estimated the grating length to be of ~4.6 mm (the length of the phase mask was 10 mm). Since we had the phase mask pitch we could obtain directly the effective refractive index and apply an iterative method to optimize the value obtained for the induced *δn*_co_. In this case the values obtained for the thermo-optic coefficients (corrected effective, core and bulk GeO_2_) were 8.48, 8.59 and 19.5 × 10^−6^ K^−1^. As can be observed the values are very close to the ones obtained previously for the moderate grating. It should be stressed that the grating temperature sensitivity (d*λ*/d*T*) depends on the fiber (with or without coating and its type), on the wavelength, and on temperature [66]. The values used (9.45 and 9.46 pm/°C) were obtained for FBGs inscribed in the SMF-28 fiber without coating at ~1.55 μm and at 20 °C. Care should be taken since the temperature sensitivity depends quadratically on temperature, a fact that sometimes seems to be ignored. We have also tested a strong FBG with a wavelength of 1608.5 nm exhibiting a sensitivity of 9.85 pm/°C (quadratic fitting between 10 and 50 °C) [67] and the results obtained were 8.50, 8.62 and 20.1 × 10^−6^ K^−1^. The temperature gauge factor *K_T_* = 1/*λ*_B_.d*λ*_B_/dT of the above gratings increased respectively, from 6.09 × 10^−6^ K^−1^ to 6.12 × 10^−6^ K^−1^, revealing the strong impact of this parameter. Note also that the reference value of 19.4 × 10^−6^ K^−1^ was obtained at room temperature in the visible region thus, assuming a similar dispersion relation for GeO_2_, as for SiO_2_, it is expected a value 6% lower at 1.5 μm. Therefore, it is instructive to measure the impact of the different parameters accuracy on the estimation of the bulk GeO_2_ thermo-optic coefficient. Starting from typical fiber parameters: *D*_co_ = 5.2–5.4 μm and Δ*n* = 5.2–5.4 × 10^−3^, differences of the order of ~4% results in relative errors of ~1%. Regarding the thermal expansion coefficient of SiO_2_ it is known that it depends on the fictive temperature and on the OH^-^ content [68,69]. Nevertheless, typical values at room temperature range from 0.40–0.55 × 10^−6^ K^−1^ [57,70,71,72]. Common values for type III silica glass at 20 °C can be considered to be (0.47 ± 0.04) × 10^−6^ K^−1^ [73,74,75,76]. For GeO_2_ at room temperature we shall consider 6.9 × 10^−6^ K^−1^ [77] (typical average value from 25 °C up to 300 °C: 7.5 × 10^−6^ K^−1^ [78,79,80,81,82]). Considering the uncertainty of 8.5% in the thermal expansion coefficient of silica glass, it impacts ~11% the value of the thermo-optic coefficient of GeO_2_ glass through Equations (12) and (13). On the other hand, the difference in determining the grating temperature sensitivity at 20 °C through a linear or quadratic fitting, results in an uncertainty of 6.3% that leads to a 100% variation in the value of the thermo-optic coefficient of GeO_2_ glass. Being aware of that fact, a ~4.5 mm weak-FBG (R < 0.1%) was inscribed in the SMF-28 fiber, where a single pulse of 3 mJ at 248 nm was used through a phase mask of 1065.38 nm. The FBG has a resonance wavelength of 1541.803 nm having, therefore, an effective refractive index of 1.447186 (differences to the effective refractive index of the fiber also arises from the strain used during the grating fabrication [18,83]). The thermal behavior of the FBG was studied from 5 °C up to 95 °C and after fitting with a second order polynomial we obtained a value of 9.454 pm/°C at 20 °C. Figure 2 shows the temperature dependence of *K*_T_ for this grating. For this weak-FBG, the former values of the thermal expansion coefficients would lead to thermo-optic coefficients (corrected effective, core and bulk GeO_2_): 8.52, 8.63 and 20.4 × 10^−6^ K^−1^, while the new values (0.47 × 10^−6^ K^−1^ and 6.9 × 10^−6^ K^−1^) lead to 8.46, 8.55 and 18.3 × 10^−6^ K^−1^. The latter corresponds to a 6% reduction going from visible to the infrared, as observed for bulk SiO_2_. The value of the effective d*n*/d*T* (without correction) is 8.20 × 10^−6^ K^−1^. Applying to the temperature dependence of the Bragg wavelength a similar analysis as the one presented in [16], that is, considering that the period of the phase mask increases linearly with temperature and the refractive index has a quadratic behavior (the reference is 20 °C), yields a value of 8.25 × 10^−6^ K^−1^ (0.6% higher). The core’s thermo-optic coefficient increases linearly with GeO_2_ concentration (mol%) at a ratio of ~0.106. Recently [67], it was suggested that the cladding of the SMF-28 fiber would have similar thermo-optic coefficients as the Suprasil glass. However, based on our results for Suprasil 3001 [8], this would lead to a thermo-optic coefficient of bulk GeO_2_ of 11.56 × 10^−6^ K^−1^, which is not correct. Therefore, the reason for the discrepancy lays in the higher values obtained for *K_T_* as a consequence of the linear fitting applied to the Bragg wavelength.

From the above calculations, our model points towards a value of 18.3 × 10^−6^ K^−1^ (20 °C and 1.55 μm) for the thermo-optic coefficient of bulk GeO_2_. In order to validate that estimative, we have used the Prod’homme equation [84,85]:(23)$$\frac{dn}{dT}=\frac{\left(n^{2}-1\right)\left(n^{2}+2\right)}{6n}\left(\phi -3\alpha \right),$$
where ϕ is the temperature coefficient of the electronic polarizability. First, we calculated ϕ for 0.546 μm by considering d*n*/d*T* = 19.4 × 10^−6^ K^−1^ [45] and the values of *n* taken from [55]. Afterwards, we estimated the thermo-optic coefficient for 1.55 μm to be 18.4 × 10^−6^ K^−1^, being in excellent agreement with our former value. Note that we have considered that ϕ is wavelength independent, although as observed for SiO_2_, it might decrease slightly with the wavelength.

As a final remark, note that different values can be found in the literature for the parameters *ν*, *p*_11_ and *p*_12_ since they may depend on temperature, wavelength and if it is a bulk or fiber glass. Therefore, in the next section we will present another approach to obtain the correction factor, ${\frac{dn_{eff}}{dT}}_{corr}$.

## 4. Effective Parameters *ν*, *p*_11_ and *p*_12_ for the SMF-28 Fiber

The correction factor will be determined by using Equation (13) through the parameters for the SMF-28 fiber. The Poisson ratio, for the fiber cladding is obtained by applying the following expression:(24)$$\nu =\frac{1-2{\left(\frac{v_{s}}{v_{L}}\right)}^{2}}{2\left(1-{\left(\frac{v_{s}}{v_{L}}\right)}^{2}\right)},$$
where *v*_S_ and *v*_L_ are the transverse and longitudinal acoustic velocities and can be determined by knowing the cladding radius [86]:(25)$$\frac{v_{s}}{R_{cl}}=59.345\pm 0.009\;m/(s.\mu m),$$



(26)
$$\frac{v_{L}}{R_{cl}}=94.463\pm 0.006\;m/(s.\mu m).$$



Thus, *ν* for SiO_2_ cladding is obtained by the ratio of the acoustic velocities yielding 0.1740 ± 0.0002 at 20 °C. To determine Pockels’ coefficients for silica cladding we followed the procedure presented in [47], that relies on the strain dependence of TE and TM polarized whispering gallery modes (WGM) resonances. The Pockels’ coefficients were determined for two wavelengths, 1.064 μm and 1.55 μm. When using the obtained values for the calculation of the photoelastic constant, *C*, we found that it would be larger at the longer wavelength, what is not correct (to be discussed below) [87,88,89]. Thus, by careful analysis of the figures in [47,90], we realized that the slopes in those figures were incidentally interchanged. Therefore, the correct values at 1.064 μm and 1.531 μm are: *p*_11_ = 0.113, *p*_12_ = 0.250 and *p*_44_ = −0.0685 and *p*_11_ = 0.130, *p*_12_ = 0.265 and *p*_44_ = −0.0676, respectively. It is interesting to note that the coefficients are essentially wavelength independent in the 3rd telecommunication window: d*p*_11_/d*λ* = 3.66 × 10^−5^ nm^−1^, d*p*_12_/d*λ* = 3.28 × 10^−5^ nm^−1^ and d*p*_44_/d*λ* = 1.93 × 10^−6^ nm^−1^. The signs of the wavelength dependence of Pockels’ coefficients compare fairly well in the visible and near infrared range [88,89]. For the sake of further comparison (we will use instead the stress-optic rotation coefficient, *g* defined below by Equation (33)), the latter value corresponds to d*g*/d*λ* = −0.069*g*/*λ* nm^−1^, which is in excellent agreement with the value of −0.069*g*/*λ* obtained for a silica fiber core-doped with 3.4 mol% GeO_2_ and B_2_O_3_ co-doped cladding, in the 1.064–1.3 μm wavelength range [91]. On the other hand, for a pure silica-core fibers and B_2_O_3_ co-doped cladding a value of −0.056*g*/*λ* nm^−1^ was obtained in the 630–880 nm [92] and for dispersion-shifted fibers (DSF) d*g*/d*λ* = −0.090*g*/*λ* nm^−1^ at 1.55 μm [93].

As far as the core is concerned, a more laborious path is required. First, it should be mentioned that the cladding diameter was not measured and the specifications of Fibercore SM1500 4.2/125 states that the cladding as a 2 μm uncertainty. Therefore, using the nominal radius of 62.5 μm and Equations (24) and (25) yields values of *v*_L_ = 5940 ± 95 m/s and *v*_S_ = 3709 ± 60 m/s, respectively. On the other hand, for the same fiber, the echo of the longitudinal and transverse acoustic waves reflecting at the cladding/coating interface repeats at a periodicity of ~21 ns and ~33 ns (with a 0.1 ns resolution) [94], leading to velocity values around 5952 m/s and 3788 m/s. Due to discrepancy, we will proceed through the analysis of the stimulated Brillouin gain spectrum (SBS). The longitudinal acoustic velocity,*ν*_L_ can be related to the Brillouin frequency shift, *f*_B_ through the following equation [95]:(27)$$f_{B}=\frac{{2n_{eff}v}_{L}}{\lambda_{p}},$$
where *n*_eff_ is the effective refractive index and *λ*_p_ the pump wavelength. We have used data corresponding to three germanium-doped silica fibers (3.65 and 8 mol% GeO_2_) [95], being one the SMF-28 fiber (3.67 mol% GeO_2_) [96,97,98]. We have also corrected the effect of the drawing tension on the Brillouin frequency shift (−42 MHz/100 g) considering a drawing tension similar to the one used in the SMF-28 fiber [99]. Table 2 summarizes the results at 20 °C.

It should be mentioned, that, we have limited the maximum value of GeO_2_ core-dopant concentration in order to calculate the effective indices through the above equations valid for weakly-guiding fibers. Care should also be taken since the Brillouin frequency shift/velocity depends on several fiber properties [100]. The extrapolated value obtained for silica glass is in good agreement with the 5990 ± 10 m/s referenced in [43,101] and it also corresponds to the value obtained for the L_04_ longitudinal acoustic mode at 20 °C (5987 m/s) [102]. The value obtained for the SMF-28 fiber is also a common accepted one (5820 m/s) [103]. Following Koyamada et al. [104] relation between longitudinal velocity and GeO_2_ concentration ([GeO_2_] < 20 mol%) we obtain:(28)$$v_{L}=5987(1-7.7\times 10^{-3}\times C_{GeO_{2}}).$$

In which concerns the transverse velocity we determine *v*_S_ in silica from Equation (23) yielding a value of 3761 m/s. Note that a value of 3764 m/s was also obtained through analysis of leaky surface acoustic waves in several Corning silica samples [105]. It is interesting to note that using these values (5987 m/s and 3761 m/s) we find *R*_cl_ = 63.4 μm being within the accuracy stated in the fiber’s specifications. It should be stressed that although in [104] it was considered the concentration in wt%, in fact it should be mol% [106]. Also, the fibers used as one of the references [107] for the Koyamada’s equations contains B_2_O_3_ in the cladding, with different concentrations, affecting the values obtained for the velocities. As a first guess, we estimated a value of 3673 m/s for *v*_S_ in the SMF-28 fiber:(29)$$v_{S}=3761(1-6.4\times 10^{-3}\times C_{GeO_{2}}),$$
and, therefore, the Poisson ratio for the SMF-28 fiber would be 0.169.

The elastic properties of materials, longitudinal and shear modulus, *M* and *G*, respectively, can be determined directly from the knowledge of the acoustic velocities:(30)$$M=\rho v_{L}^{2},$$



(31)
$$G=\rho v_{S}^{2}.$$



On the other hand, the Young’s modulus, *E*, can be related to *M* through the Poisson’s ratio *ν*:(32)$$\frac{M}{E}=\frac{1-\nu }{\left(1+\nu \right)\left(1-2\nu \right)}.$$

Therefore, for SiO_2_ we get *M* = 78.86 GPa, *E* = 73.08 GPa, and *G* = 31.12 GPa. For the SMF-28 fiber *M* = 76.36 GPa and *E* can be estimated by knowing the effect of GeO_2_ concentration on Young’s modulus [78]. A decrease of ~0.35 GPa/mol% was found for GeO_2_ concentrations up to 4 mol%, although the temperature and density should be corrected [108]. On the other hand, from results presented in [109] a value of −0.4 GPa/mol% can be determined. Therefore, we estimate *E* = 71.61 GPa for the SMF-28 fiber which is in excellent agreement with the value measured for an SMF without coating, *E* = 71.63 ± 0.43 [110]. A lower value was obtained for the SMF-28e (70.05 ± 0.34) [111], however it requires a precise measurement of the fiber cladding diameter which was not performed. Moreover, by applying the additivity model and by using data related to pure bulk SiO_2_ and GeO_2_ from [112] results a value *E* = 71.65 GPa which, once again, validates our result. For the sake of completion, the model can be improved by considering other factors such as the dissociation energy and ionic radius [113,114,115]. From the values of *E* and *M* for the SMF-28 fiber results *ν* = 0.1612, also in accordance to [109] and thus *v*_S_ = 3698 m/s. Therefore, Equation (29) should be corrected to be
(29*)$$v_{S}=3761(1-4.6\times 10^{-3}\times C_{GeO_{2}}).$$

It should be highlighted that the obtained Poisson’s ratio is lower for GeO_2_ doped silica glass fibers which also agrees with [107], but it is in contradiction to what is expected by applying the additivity model to bulk glasses [116]. Following Equations (28) and (29*) the Poisson ratio decreases with GeO_2_ concentration increase ([GeO_2_] < 20 mol%) according to the expression: *ν* = −6.6945 × 10^−5^[GeO_2_]^2^ − 3.1851 × 10^−3^[GeO_2_] + 0.1740. We are aware that the results obtained depend on the initial values, but by following the existing interconnection between several parameters allows us to validate the results. We shall now work, with Pockels’ coefficients (*p*_11_, *p*_12_ and *p*_44_), stress-optic rotation coefficient (*g*) and photoelastic constants (*C* = *C*_1_ − *C*_2_). The coefficient *g* can be determined through twist/rotation measurements and is related to *p*_44_ through the equation:(33)$$g=-n^{2}p_{44},$$
and
(34)$$C=C_{1}-C_{2}=\frac{n^{3}p_{44}}{2G},$$
where the photoelastic constants, longitudinal *C*_1_ and transverse *C*_2_ are defined as:(35)$$C_{1}=\frac{n^{3}\left(p_{11}-2\nu p_{12}\right)}{2E},$$



(36)
$$C_{2}=\frac{n^{3}\left(p_{12}-\nu \left(p_{11}{+p}_{12}\right)\right)}{2E}.$$



Note that *g* is related to *C* and *p*_44_ = (*p*_11_ − *p*_12_)/2. Through the use of whispering gallery modes [47] we achieved a value of *g* = 0.141 at 1.55 μm. In general, values for SMF range from 0.140 to 0.144 [117,118,119]. We also found a value for the SMF-28 fiber [120] that may be 0.139 ± 0.002, since the slope taken from Figure 3 of that paper is at least 69.3 × 10^−3^ and not 63.9 × 10^−3^ as stated in the document. Thus, by applying Equation (33) we obtain *p*_44_ = −0.0662 and by using the value of 0.205 ± 0.004 for the effective strain-optic coefficient *p*_eff_ reported in the strain measurements of FBGs [121,122]:(37)$$p_{eff}=\frac{EC_{2}}{n}=\frac{n_{eff}^{2}}{2}\left(p_{12}-\nu \left(p_{11}{+p}_{12}\right)\right),$$
we found for the SMF-28 fiber, *p*_11_ = 0.1251, *p*_12_ = 0.2575. Table 3 summarizes the results for SiO_2_ and the SMF-28 fiber. Inserting the values in Equation (13) results a value for the correction factor that differs only in the fourth decimal place when compared to the initial one. Therefore, we conclude that the major factor that impacts the value of the thermo-optic coefficient is the temperature sensitivity of the FBGs.

It is instructive to note that if we consider for the SMF-28 fiber a value of *g* = 0.141 [119], it would result in *p*_44_ = −0.0672 and *C* = −3.30 × 10^−12^ Pa^−1^. Thus, due to the uncertainty in all calculations, the best we can say is that the photoelastic constant for the SMF-28 fiber is very close to the one obtained for pure silica cladding fiber, which is also in excellent agreement with previous results [123,124]. Sinha [125] proposed an expression for the dispersion of the photoelastic constant of fused silica by fitting data from Jog and Krishnan [87] and from Primak and Post [126].
(38)$$C_{\lambda }=C_{\lambda_{0}}\frac{n_{\lambda_{0}}}{n_{\lambda }}\frac{\lambda^{2}}{\lambda_{0}^{2}}\frac{\left(\lambda_{0}^{2}-\lambda_{1}^{2}\right)}{\left(\lambda^{2}-\lambda_{1}^{2}\right)}\frac{\left(\lambda^{2}-\lambda_{2}^{2}\right)}{\left(\lambda_{0}^{2}-\lambda_{2}^{2}\right)},$$
where *λ*_1_ = 0.1215 μm and *λ*_2_ = 6.900 μm and the normalization was considered at 0.541 μm to be *C* = 3.63 and 3.56 (absolute value in brewster = 10^−12^ Pa^−1^) for each data set, respectively. Figure 3 shows the dispersion of the photoelastic constants for bulk fused silica (Jog and Primak), for the SMF-28 fiber (cladding and core) estimated using the Wemple-DiDomenico model (WDM) [127] and for low concentration GeO_2_-doped silica fiber [124]. The Pockels’ coefficients can be determined by using the following expression from WDM:(39)$$p_{ij}=\frac{2}{E_{d}}{\left(1-\frac{1}{{n(\lambda )}^{2}}\right)}^{2}D_{ij}\left[1+K_{ij}{\left(\frac{1.23498}{E_{0}\lambda }\right)}^{2}\right],$$
where *E*_0_ is the average electronic energy gap and *E*_d_ is the electronic oscillator strength as discussed in [8] and *D*_ij_ and *K*_ij_ are fitting parameters related to changes in the former parameters due to strain [89]. For the cladding we have used data from [8] and the Pockels’ coefficients obtained at 1.064 and 1.55 μm. For the core, the estimative was performed by using data at 1.55 μm and for 1.4 μm we have assumed the same wavelength dependence for the Pockels’ coefficients as determined for the cladding. For further analysis, discussions presented in [8,128] should be followed. Table 4 summarizes the different parameters obtained for the cladding and core of the SMF-28 fiber.

It can be observed in Figure 3 that the photoelastic constants are lower for optical fibers when compared to bulk samples and that the values for the core region (doped with low GeO_2_ concentrations, less than for the SMF-28 fiber) are lower than the ones obtained for the cladding. From the experimental values and due to uncertainty [124] it is not possible to clearly state that GeO_2_ increases/decreases the value of the photoelastic constant despite it seems that it affects *g* [119] and, therefore, *p*_44_ and ultimately *C*.

The temperature dependence of Pockels’ coefficients can be obtained from the temperature derivative of Equation (34) and from SBS spectrum [106]:(40)$$g_{0}=\frac{{2\pi n}^{7}p_{12}^{2}}{c\lambda_{p}^{2}\rho v_{L\Delta f}},$$
where *c* is the light speed, *λ*_p_ is the pump wavelength, Δ*f* is the spectral width and *g*_0_ the intensity. Considering that the product intensity-spectral width is temperature independent [106], results:(41)$$v_{L}\alpha n^{7}p_{12}^{2},$$
and therefore,
(42)$$\frac{1}{v_{L}}\frac{dv_{L}}{dT}=7\frac{1}{n}\frac{dn}{dT}+2\frac{1}{p_{12}}\frac{dp_{12}}{dT}.$$

Since the temperature derivative of the acoustic velocity is ~0.57 [86] and that the normalized thermo-optic coefficient equals 5.65 × 10^−6^ K^−1^ [8], yields d*p*_12_/d*T* = 0.74 × 10^−5^ K^−1^.

From Equations (31) and (34), results:(43)$$\frac{1}{p_{44}}\frac{dp_{44}}{dT}=2\frac{1}{v_{S}}\frac{dv_{S}}{dT}+\frac{1}{C}\frac{dC}{dT}-3\frac{1}{n}\frac{dn}{dT},$$
taking into consideration that d*v*_S_/d*T* = 0.22 [86] and that (1/*C*)d*C*/d*T* = 1.34 × 10^−4^ K^−1^ [91], yields d*p*_44_/d*T* = −1.57 × 10^−5^ K^−1^. Finally, from the relation between the Pockels’ coefficients, d*p*_12_/d*T* = −2.40 × 10^−5^ K^−1^. The temperature dependence of the Poisson ratio is d*ν*/d*T* = 3.76 × 10^−5^ K^−1^ [86]. Due to the weak dependence on temperature exhibited by the Pockels’ coefficients and Poisson ratio, the correction factor is essentially dominated by the difference in the thermal expansion coefficients of the core and cladding. Figure 4 shows the thermal expansion coefficient for pure silica and the SMF-28 fiber. As can be observed the difference between the two curves decreases as the temperature decreases and, consequently, the correction factor also decreases. The thermal expansion coefficient for the SMF-28 fiber was obtained through the use of the additivity model (<4 mol% GeO_2_) [57,79,80] where for GeO_2_ glass [81,129] we have used data from [82,130] but with fix values at very low temperatures [131], 293 K [77] and 473 K [132]. Following this procedure, data was fitted with an equation similar to the one used by Okaji et al. [71] for SiO_2_ glass being the coefficients 1.27, 82.42, 1.23, 8.85 and 522.8, respectively. As a final remark, it should be mentioned that without the correction factor, the effective thermo-optic coefficient for the core (SMF-28 fiber) and cladding would be essentially the same. Figure 5 was obtained by using Equations (6), (7) and (12), the values adjusted of *K_T_* from [62] and the temperature dependence of the SiO_2_ from [8]. An estimative for the temperature dependence of the thermo-optic coefficient is presented in Figure 6, where we have assumed a linear dependence on temperature for Pockels’ coefficients and Poisson ratio [86].

Other potential techniques to determine the thermo-optic coefficient of an optical fiber are based, for instance, on Fabry-Perot interferometers (FPI) [133,134,135,136,137], Rayleigh backscattering [138,139,140] and optoelectronic oscillations [141,142]. FPI is the most used approach and thus, for the sake of comparison we estimated from [134] a value of 8.22 × 10^−6^ K^−1^ for the effective thermo-optic coefficient of a standard fiber, being therefore in excellent agreement (with the value, without correction, obtained in the previous section). On the other hand, following the procedure presented in [133] the values ranged from 8.10 × 10^−6^ K^−1^ (average heating cycles) up to 8.70 × 10^−6^ K^−1^ (first heating up). The reasons for the discrepancy are related with two facts: first, the reference temperature, 20 °C, is the lower limit of the temperature interval of the experiment causing uncertainties to the derivative of the fitting equation; second, heating successively above 600 °C makes irreversible changes to the glass structure which affects the temperature sensitivity and, consequently, the thermo-optic coefficient.

## 5. Conclusions

We have derived an expression that allows to determine the thermo-optic coefficient of weakly guiding fibers. Our analysis was based on FBG although it can also be used for FPI and Rayleigh scattering. We concluded that the thermo-optic coefficient (effective) of the SMF-28 fiber and of the cladding are essentially the same if one does not consider the correction factor. We estimated the thermo-optic coefficient of the bulk GeO_2_ glass from 200 K up to 300 K being 18.3 × 10^−6^ K^−1^ at 293 K and 1.55 μm. We obtained an expression for the temperature dependence of the thermal expansion coefficient of GeO_2_-doped silica fibers. Expressions for the transverse and longitudinal acoustic velocities as a function of GeO_2_ concentration were also presented. We have determined values for the Poisson ratio, the Pockels’ coefficients and photoelastic constant for the SMF-28 fiber. We have also discussed the dispersion and temperature dependence of Pockels’ coefficients. We are aware of the uncertainty of some values used, however this paper presents the relations between the different parameters that allow a straightforward correction, if required. Therefore, currently we are investigating the temperature sensitivity of FBGs inscribed, with femtosecond and UV laser radiation, in fibers with different GeO_2_ concentration and we are also researching the effect of the Bragg wavelength, hydrogen loading and reflectivity, and the results will be published elsewhere.

## Figures and Tables

**Figure 1 sensors-24-04857-f001:**
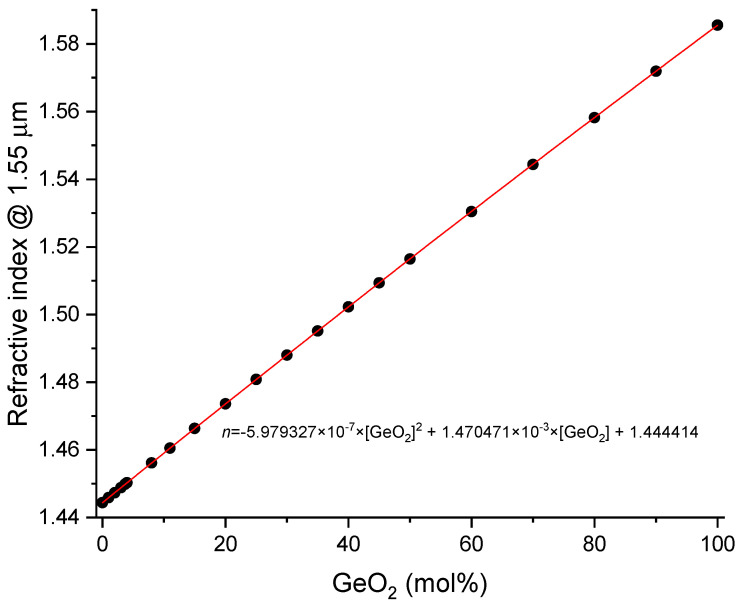
Refractive index of binary SiO_2_-GeO_2_ glasses at room temperature and 1.55 μm.

**Figure 2 sensors-24-04857-f002:**
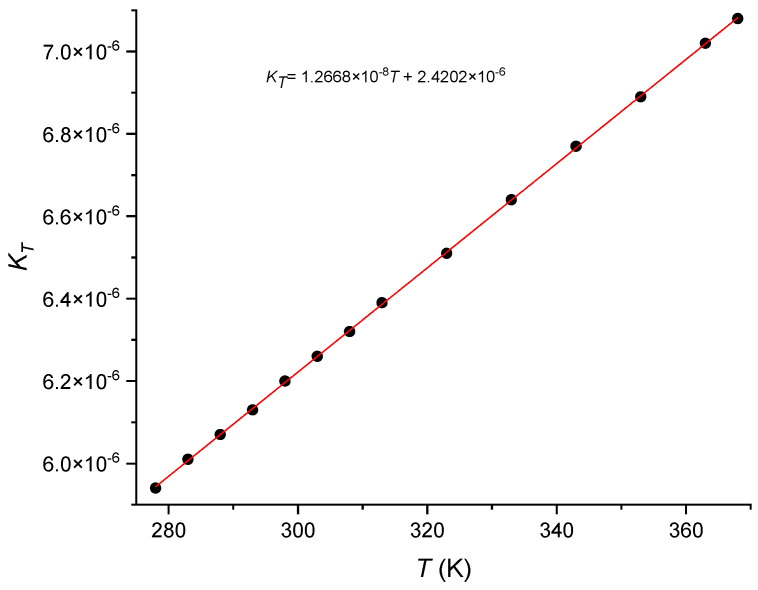
Temperature gauge factor of a weak-FBG.

**Figure 3 sensors-24-04857-f003:**
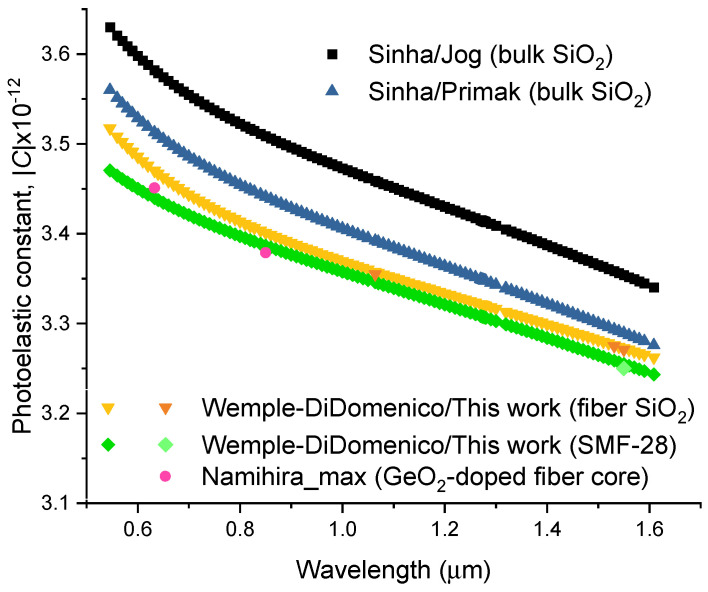
Dispersion of the photoelastic constant for SiO_2_ and SiO_2_-GeO_2_ glasses.

**Figure 4 sensors-24-04857-f004:**
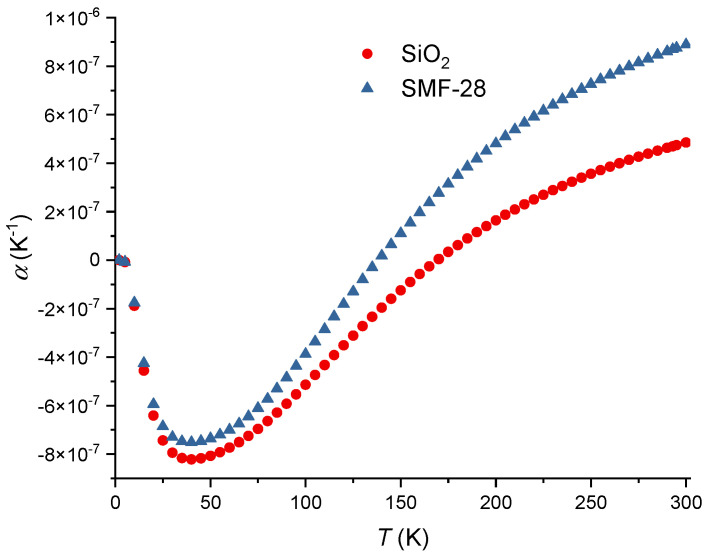
Thermal expansion coefficients for SiO_2_ and SMF-28 fiber.

**Figure 5 sensors-24-04857-f005:**
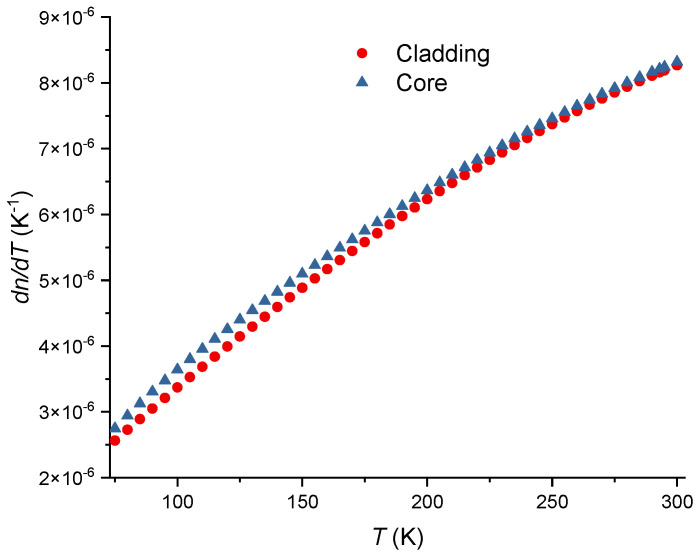
Thermo-optic coefficients for the cladding and core of the SMF-28 fiber.

**Figure 6 sensors-24-04857-f006:**
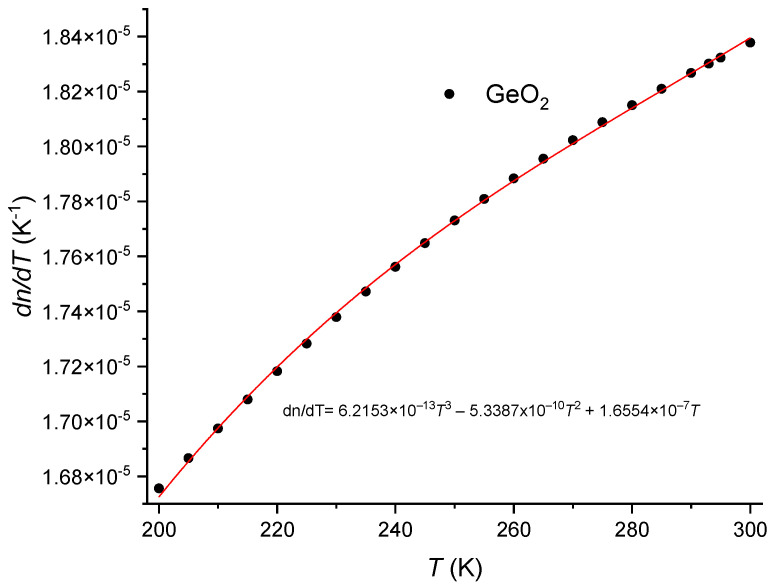
Thermo-optic coefficient for bulk GeO_2_ glass.

**Table 1 sensors-24-04857-t001:** Physical parameters.

Material	*M* (g/mol)	*ρ* (kg/m^3^)	*α* (×10^−6^ K^−1^)	*p* _11_	*p* _12_	*ν*
SiO_2_	60.08	2200	0.45	0.121	0.270	0.170
GeO_2_	104.64	3650	7.7	0.130	0.288	0.212

**Table 2 sensors-24-04857-t002:** SBS and acoustic velocity.

GeO_2_ Concentration (mol%)	Brillouin Frequency, *f*_B_ (GHz)	Longitudinal Velocity, *v*_L_ (m/s)
0	11.143 (extrapolated)	5986.5 (extrapolated)
3.65	10.872	5819.5
3.67	10.863	5818.5
8.0	10.542	5620.3

**Table 3 sensors-24-04857-t003:** Physical parameters calculated for SiO_2_ and the SMF-28 fiber @1.55 μm.

	SiO_2_	SMF-28 (3.67 mol% GeO_2_)
*n*_cl_ or *n*_eff_	1.444414	1.446973
*p* _11_	0.130	0.125
*p* _12_	0.266	0.258
*p* _44_	−0.0676	−0.0662
*g*	0.141	0.139
*C*_1_ (Pa^−1^)	7.81 × 10^−13^	8.90 × 10^−13^
*C*_2_ (Pa^−1^)	4.06 × 10^−12^	4.14 × 10^−12^
*C* (Pa^−1^)	−3.27 × 10^−12^	−3.25 × 10^−12^
*M* (GPa)	78.86	76.36
*E* (GPa)	73.08	71.63
*G* (GPa)	31.12	30.83
*ν*	0.174	0.161

**Table 4 sensors-24-04857-t004:** WDM’s parameters calculated for SiO_2_ and the SMF-28 fiber.

	SiO_2_	SMF-28 (3.67 mol% GeO_2_)
*E* _d_	14.56	13.90
*E* _0_	13.24	12.505
*D* _11_	−119.44	−166.43
*K* _11_	−1.0330	−1.0233
*D* _12_	−115.96	−165.43
*K* _12_	−1.0653	−1.0439

## Data Availability

The data segments can be obtained by contacting the corresponding author.

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
