# Peer review of "Temperature Dependence of the Thermo-Optic Coefficient of GeO2-Doped Silica Glass Fiber"

_sensors, 2024, doi:10.3390/s24154857_

Round 1

Reviewer 1 Report

Comments and Suggestions for Authors

In this paper, an expression that allows the determination of the thermo-optic coefficient of weakly-guiding germanium-doped silica fibers, based on the thermal behavior of optical fiber devices was derived. The temperature dependence of the thermal expansion coefficient of GeO2 was estimated and the dispersion and temperature dependence of Pockels’ coefficients was discussed. The authors have presented expressions for the dependence of the longitudinal and transverse acoustic velocities on the GeO2 concentration used to calculate the Poisson ratio, and discussed the dispersion of the photoelastic constant. Finally, an estimate for the temperature dependence of the thermo-optic coefficient of bulk GeO2 glass is presented for the 200-300 K temperature range.

The authors did a good job of this work. However, there are still some issues in the article that need to be further clarified. I would think a minor revision will be necessary to improve the quality of this work before it can be accepted. Detailed comments can be found below.

1. Why are the photoelastic constants of smf-28 (3.67 mol% GeO2) and pure quartz clad fiber very close to each other after doping GeO2, will the photoelastic coefficients of other doping ratios be similar?

2. Can the method of measuring the thermo-Optic Coefficient in this paper be extended to other kinds of optical fibers? How is the estimated effect?

3. Please note formatting specifications, such as the subscript of GeO2. Please check the full text carefully and revise it.

Comments on the Quality of English Language

The English level in the article is basically satisfactory.

Reviewer 2 Report

Comments and Suggestions for Authors

This work discusses an expression that allows to determine the thermo-optic coefficient of weakly guiding fibers. It is shown that the developed framework possesses an immense potential to be employed for Fabry-Perot interferometers and Rayleigh scattering principles. It is also claimed that the thermo-optic coefficient (effective) of the SMF-28 fiber and the cladding are essentially the same regardless of the correction factor. This was complemented by quantitative analysis, and the thermo-optical coefficient of the bulk GeO2 glass at different temperatures was assessed. The manuscript contains interesting results and adds novel insights into the field. The work can be considered for publishing after addressing some concerns that are listed below:

1) While the impact of temperature was studied, the effect of pressure should also be considered or discussed. 

2) How the proposed framework can be extended to other materials and operating wavelengths? 

Comments on the Quality of English Language

There are some grammatical mistakes, and the work needs to be polished carefully. 

Reviewer 3 Report

Comments and Suggestions for Authors

The author carefully study the thermo-optic coefficient of GeO2-doped silica in optical fibers. By a systematic consideration of the most important physical parameters, the author obtains an expression of the thermo-optic coefficient in terms of the temperature. The topic is interesting, and the manuscript is well-written, tables and figures seem adequate and section structure make sense. In the opinion of this reviewer, minor edits must be addressed before publication in Sensors.

1.-Please define Dco, nco and ncl of equations (1) and (2) even if in general are well-known variables.

2.- Can the author address the difference between the presented approach with the construction of the temperature-dependent Sellmeier equation such as in: 10.1088/1464-4258/4/2/309

3.- Although the scare literature on thermo-optic coefficient of GeO2 doped silica, in the opinion of this reviewer, some extra references can be included such as: 10.4028/www.scientific.net/MSF.514-516.369, 10.1364/OL.27.001297, 10.1088/1464-4258/4/2/309.

4.- Please consider including a brief discussion about the potential applications that benefit from the results of the presented manuscript.

Comments on the Quality of English Language

Some typos are found along the manuscript. For example:

-An unnecessary space in the 7th line of section 2. (A long instead Along).

-In the fourth line of page 9 one reads “e” where an “and” should be read.

-A period is missing in line 6 of the conclusion section.
